

# Semantic visual simultaneous localization and mapping (SLAM) using deep learning for dynamic scenes

Xiao Ya Zhang[1], Abdul Hadi Abd Rahman[1] and Faizan Qamar[2]

[1] Center for Artificial Intelligence Technology, Universiti Kebangsaan Malaysia, Bangi, Malaysia
[2] Center for Cyber Security, Universiti Kebangsaan Malaysia, Bangi, Malaysia

## ABSTRACT

Simultaneous localization and mapping (SLAM) is a fundamental problem in robotics and computer vision. It involves the task of a robot or an autonomous system navigating an unknown environment, simultaneously creating a map of the surroundings, and accurately estimating its position within that map. While significant progress has been made in SLAM over the years, challenges still need to be addressed. One prominent issue is robustness and accuracy in dynamic environments, which can cause uncertainties and errors in the estimation process. Traditional methods using temporal information to differentiate static and dynamic objects have limitations in accuracy and applicability. Nowadays, many research trends have leaned towards utilizing deep learning-based methods which leverage the capabilities to handle dynamic objects, semantic segmentation, and motion estimation, aiming to improve accuracy and adaptability in complex scenes. This article proposed an approach to enhance monocular visual odometry's robustness and precision in dynamic environments. An enhanced algorithm using the semantic segmentation algorithm DeeplabV3+ is used to identify dynamic objects in the image and then apply the motion consistency check to remove feature points belonging to dynamic objects. The remaining static feature points are then used for feature matching and pose estimation based on ORB-SLAM2 using the Technical University of Munich (TUM) dataset. Experimental results show that our method outperforms traditional visual odometry methods in accuracy and robustness, especially in dynamic environments. By eliminating the influence of moving objects, our method improves the accuracy and robustness of visual odometry in dynamic environments. Compared to the traditional ORB-SLAM2, the results show that the system significantly reduces the absolute trajectory error and the relative pose error in dynamic scenes. Our approach has significantly improved the accuracy and robustness of the SLAM system's pose estimation.

# INTRODUCTION

Simultaneous Localization and Mapping (SLAM) (*Cadena et al., 2016*; *Siah, Abdullah & Sahran, 2013*) is a technology that simultaneously estimates the robot's trajectory and the environment's map. SLAM is mainly used in autonomous robot navigation (*Mallios et al.,*

Corresponding author
Abdul Hadi Abd Rahman,
abdulhadi@ukm.edu.my

*2010*; *Raibail et al., 2022*), unmanned vehicles (*Sun et al., 2018*; *Pavel, Tan & Abdullah, 2022*), virtual and augmented reality (*Kuswadi et al., 2018*), indoor service robots, and outdoor autonomous driving robots (*Bresson et al., 2017*; *Azmi et al., 2019*). In SLAM technology, visual SLAM (*Razali, Faudzi & Shamsudin, 2022*) is a technology that uses single or multiple cameras for real-time 3D environment reconstruction and camera positioning. Compared with others, visual SLAM has the following advantages: (i) the cost of the camera is low; (ii) the camera can obtain higher resolution when acquiring image information with more detailed texture information; (iii) more accurate feature matching and pose estimation results can be obtained; and (iv) the camera acquires images very quickly, allowing visual SLAM to operate in real-time scenes. VSLAM has a wide range of applications in fields such as robot navigation and localization, mapping, obstacle avoidance, unmanned aerial vehicles, building and indoor navigation, industrial automation, and service robots. With its real-time positioning and environmental perception capabilities, VSLAM enables robots and unmanned systems to intelligently navigate and operate in unknown or dynamic environments. As technology continues to advance, the improvement of computational power, sensor technology, and algorithm optimization will continuously enhance the performance and reliability of VSLAM. As a core technology, VSLAM will continue to play a crucial role in the fields of robotics and computer vision, providing smarter and more autonomous solutions for various application scenarios. In the past few decades, the problem of visual SLAM has received much attention, and some advanced visual SLAM algorithms have achieved satisfactory performance (*Zhang et al., 2022a*, *2022b*; *Zhao et al., 2022*; *Rahman et al., 2019*).

*Klein & Murray (2008)* proposed a keyframe-based visual SLAM algorithm called PTAM (Parallel Tracking and Mapping). The algorithm divides the localization and mapping tasks into two parallel threads, incorporates a keyframe mechanism, and employs bundle adjustment (BA) for nonlinear optimization processing. These techniques significantly reduce the image processing volume, enhance the system's efficiency, and preserve adequate visual environmental information. In 2015, *Mur-Artal, Montiel & Tardos (2015)* proposed the classic ORB-SLAM algorithm based on a monocular camera. The algorithm borrowed ideas from the PTAM algorithm and combined ORB features with a bag-of-words (BoW) based on the loop-closure detection method. The SLAM system was divided into three parts: tracking, local mapping, and loop-closure detection, and ran with three synchronized threads. This system has the advantages of high localization accuracy making it possible to run on low-performance embedded devices. In 2017, the team improved ORB-SLAM and proposed the ORB-SLAM2 (*Mur-Artal & Tardos, 2017*) system, which supports stereo and RGB-D camera modes and is currently one of the most widely used visual SLAM systems.

However, with the development of deep learning methods in image processing, segmentation algorithms based on deep learning have significant advantages in both accuracy and speed compared to traditional clustering segmentation algorithms at the region level. Currently, most deep learning-based semantic segmentation methods are based on fully convolution networks (FCN) (*Long, Shelhamer & Darrell, 2015*), which convent well-known classification models, including AlexNet, Vgg-16, GoogleNet (*Szegedy*

*et al., 2015*), and ResNet (*He et al., 2016*), into fully convolutional models, achieving end-to-end training for semantic segmentation problems on convolutional neural networks (CNNs). Based on this, *Ronneberger, Fischer & Brox (2015)* proposed that the U-Net network concatenates the feature maps of the encoder onto the up-sampled feature maps of the corresponding decoder, allowing the decoder to learn relevant features lost during encoder pooling at each stage. This method can handle large-resolution images and can be trained with small datasets but cannot handle scale variation well. In 2014, *Chen et al. (2014)* designed the DeeplabV1 model, which converts the fully connected layers of VGG-16 into convolutional layers and uses dilated convolutions to expand the receptive field without increasing the parameters. Based on this, the DeeplabV2 (*Chen et al., 2017a*) model replaces the VGG-16 backbone with ResNet and introduces the Atrous Spatial Pyramid Pooling (ASPP) module to obtain multi-scale features. Finally, the boundary information is refined by the Dense Conditional Random Field (Dense CRF). The subsequent DeeplabV3 (*Chen et al., 2017b*) model removes the Dense CRF and uses dilated convolution modules of different scales to obtain multi-scale contextual information. In 2018, the team designed the DeeplabV3+ (*Chen et al., 2018*) network model based on the DeeplabV3 model. The model consists of an encoder and a decoder, where the structure of the encoder is the DeeplabV3 network, and the Xception network is used as the feature extraction network. The decoder uses bilinear up-sampling and information from the encoder stage to help restore the details and spatial dimensions of the target.

## BACKGROUND

Most existing visual SLAM algorithms are based on static environment assumptions, which have low accuracy and poor robustness in dynamic environments. Moreover, the maps they create are usually based on geometric information, such as landmark-based maps and point cloud maps. Therefore, they cannot provide a high-level understanding of the surrounding environment. To solve this problem, many scholars have proposed SLAM methods for dynamic environments. The detection methods of dynamic features in scenes can be divided into two categories: dynamic feature detection, which relies only on geometric information, and dynamic feature detection, which relies only on semantic information.

In the methods that rely on geometric information, some common methods for handling dynamic scenes include filtering, graph optimization, data association, and motion estimation methods. The Extended Kalman Filter (EKF) (*Viset, Helmons & Kok, 2022*) and Unscented Kalman Filter (UKF) are the most commonly used filtering methods. These algorithm model dynamic targets as state variables and use observed data for state estimation and prediction, enabling tracking of dynamic objects and map updates. Graph optimization methods (*Chen et al., 2022*; *Jia et al., 2022*) construct a graph model representing sensor measurements and state variables as modes. Optimization algorithms are then used to minimize the error function. in the graph, dynamic objects are typically modeled as uncertain nodes connected to nodes representing the static environment. The optimization algorithm can simultaneously estimate camera poses, map topology, and dynamic object states. Data association methods (*Rakai et al., 2022*) involve

nearest-neighbor matching and data association. These methods use the motion models of objects and sensor measurements to determine whether they belong to the same object. Motion estimation methods analyze pixel differences or feature matches between consecutive frames to detect dynamic objects and utilize their motion information for tracking and mapping. Common techniques include optical flow or feature-based methods such as SURF (*Bay et al., 2008*) and SIFT (*Rublee et al., 2011*), which extract and match features.

In semantic information, objects that can move are often considered dynamic objects based on human experience and attempts. SOLO-SLAM (*Sun et al., 2022*) system is based on the ORB-SLAM3 (*Campos et al., 2021*) algorithm, and it uses YOLO (*Wu et al., 2022*) target detection networks to obtain semantic information and remove unstable features on moving objects and improves upon it by introducing semantic threads and a new dynamic point filtering strategy. By parallelizing the semantic and SLAM threads, the system enhances the real-time performance of SLAM systems. The system also incorporates a combination of dynamic regional degree and geometric constraints to enhance the filtering effect for dynamic points. TwistSLAM (*Gonzalez et al., 2022*) algorithm is based on the ORB-SLAM2. The system simultaneously estimates the camera pose and the motion of moving objects in the scene, where the map constrains the movement of objects. During mapping, semantic information is used to construct cluster maps corresponding to objects in the scene. Once the cluster maps are built, the pose estimation can be performed using only the static clusters (such as roads and buildings). The dynamic clusters can be tracked and constrained by their velocity changes. DynaSLAM (*Bescos et al., 2018*) is based on ORB-SLAM2, which incorporates dynamic object detection and background inpainting capabilities. It utilizes multi-view geometry and deep learning methods to detect dynamic objects and generates a static scene map based on this information. Then, the input frames are processed to fill the regions occluded by dynamic objects, effectively restoring the background.

The common deep learning networks used for SLAM and semantic analysis are FCN, SegNet, and DeepLab. FCN (*Villa et al., 2018*) is primarily used for semantic segmentation tasks. It replaces the full connected layers in traditional CNNs with convolutional layers, allowing the network to take images of any size as input and produce semantic segmentation results of the same size. However, FCN tends to preserve less fine-grained details, leading to less precise segmentation boundaries. In the SegNet (*Badrinarayanan, Kendall & Cipolla, 2017*) framework, during the up sampling process, it can partially reconstruct the original image, but still loses a significant amount of fine-gained information due to the presence of pooling layers. Although pooling operations can enlarge the receptive field, they also filter and discard a large amount of fine-grained features. Consequently, the DeepLab (*Chen et al., 2017a*) model emerged. DeepLab discards pooling layers, avoiding data loss during the pooling process, and adopts dilated convolutions to increase the receptive field. Detailed convolutions contain two pieces of information: the size of the convolution kernel and the dilation rate, which represents the extent of expansion. Additionally, DeepLab incorporates a conditional random field (CRF) (*Quattoni, Collins & Darrell, 2004*) to improve segmentation accuracy. The CRF layer is

generally placed at the end of the network, imposing certain constraints on the final predicted results to ensure consistency in object label predictions within the images.

DeeplabV3+ (*Chen et al., 2018*) is the Deep Lab series's latest improved model version. It adopts Atrous Spatial Pyramid Pooling (ASPP) to capture multi-scale contexture information by analyzing images at different scales using dilated convolutions with varying rates. This enables the model to understand objects of different sizes and capture fine-grained details. The model follows an encoder-decoder architecture, where the encoder extracts high-level features from the input image, and the decoder upsamples these features to generate the final segmentation map. In the decoder, depth-wise separable convolutions are used to reduce computational complexity, and skip connections are employed to combine low-level and high-level features, improving object boundary delineation. Additionally, DeepLabV3+ introduces image-level feature integration through global pooling, further enhancing the model's understanding of the global context.

This article presents a method that combines semantic segmentation technology with motion consistency detection algorithms to eliminate dynamic object features and reduce the errors in pose estimation in SLAM. By leveraging the prior information from semantic segmentation and geometric data, the method effectively diminishes the impact of dynamic objects. Moreover, in our research, we incorporated DeeplabV3+ into the feature extraction stage of our dynamic SLAM system. Specifically, we used DeeplabV3+ to identify dynamic objects in the image and then used a motion consistency check algorithm to remove further feature points belonging to dynamic objects. This approach helped improve our SLAM system's robustness in dynamic environments.

## ALGORITHM FRAMEWORK

The framework used in this article is shown in Fig. 1. First, ORB features are extracted from the input image frame. Then, it uses DeeplabV3+ for semantic segmentation to identify dynamic objects in the scene. A motion consistency check algorithm removes ORB features belonging to dynamic objects. Then use the remaining static ORB features for feature matching. The camera pose is then estimated using the matched ORB features. Loop closures are then detected to correct for accumulated drift errors in the estimated trajectory. Finally, the estimated map is optimized using a bundle adjustment algorithm.

SLAM method incorporates several key steps to improve the accuracy and robustness of pose estimation in dynamic environments. Initially, ORB features, which are distinctive key points, are extracted from the input image frames. Semantic segmentation using DeeplabV3+ is then employed to identify and label dynamic objects in the scene. A motion consistency check algorithm is subsequently applied to remove the ORB feature associated with dynamic objects, effectively mitigating their impact on pose estimation. Feature matching is performed using the remaining static ORB features, establishing correspondences between consecutive frames. The matched features are then utilized to estimate the camera's pose through optimization techniques and triangulation (*Hartley & Sturm, 1997*) which recovers the depth information of the 3D point P, thereby estimating the actual position of the feature point in 3D space. This provides crucial map information and camera pose estimation for the SLAM system. Finally, loop closure detection is

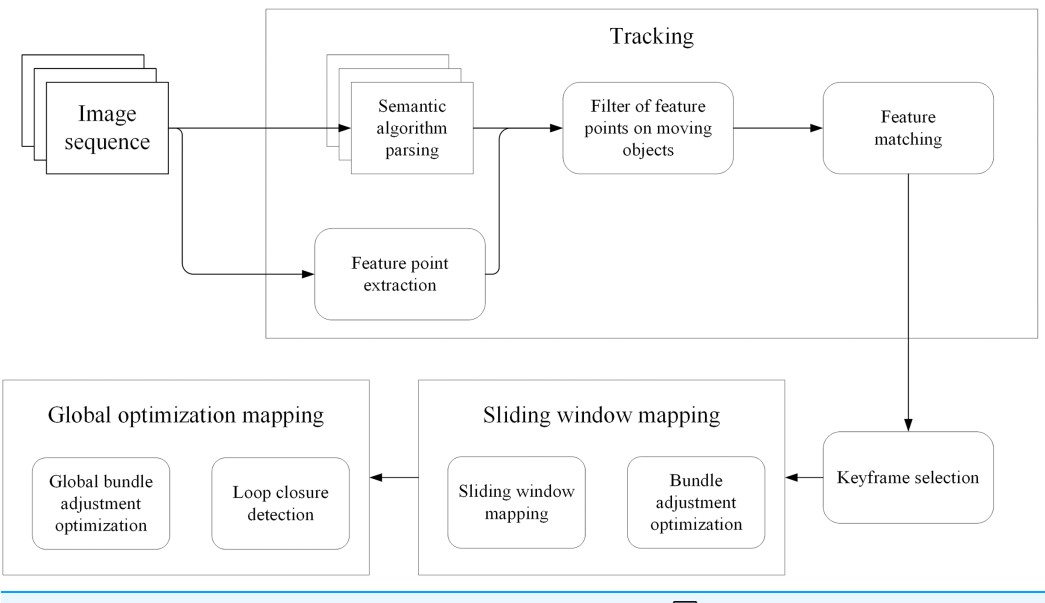

**Figure 1** **The framework of the whole system.**

implemented to identify previously visited locations, enabling the correction of accumulated drift errors in the estimated trajectory. Finally, the estimated map is optimized using a bundle adjustment algorithm, refining its accuracy and consistency. These steps together contribute to reliable and accurate simultaneous localization and mapping in dynamic environments.

## DeeplabV3+

The DeeplabV3+ network is composed of two parts: encoding and decoding modules. The encoding module comprises improved Xception network (*Chen et al., 2018*) and ASPP (*Veeravasarapu, Rothkopf & Visvanathan, 2017*). As shown in Fig. 2, the training samples extract features through the Xception network, obtain multi-scale information through ASPP and aggregate global features, and finally output the feature map with deep features by 1 × 1 convolution. Make bilinear up-sampling for the feature map and 1 × 1 convolution for the same resolution shallow features corresponding to the Xception network. Finally, the shallow features and deep features are connected by convolution fusion, and the multi-scale features are bilinearly up-sampled to achieve classification prediction.

### *Xception*

The Xception network (*Yu et al., 2022*) framework is divided into three parts: the entry flow, intermediate flow, and exit flow. The entry stream is used to sample the input image to reduce the space size, while the middle stream is used to learn the correlation relationship and continuously optimize the features. The exit stream sorts the features to obtain a rough score map.

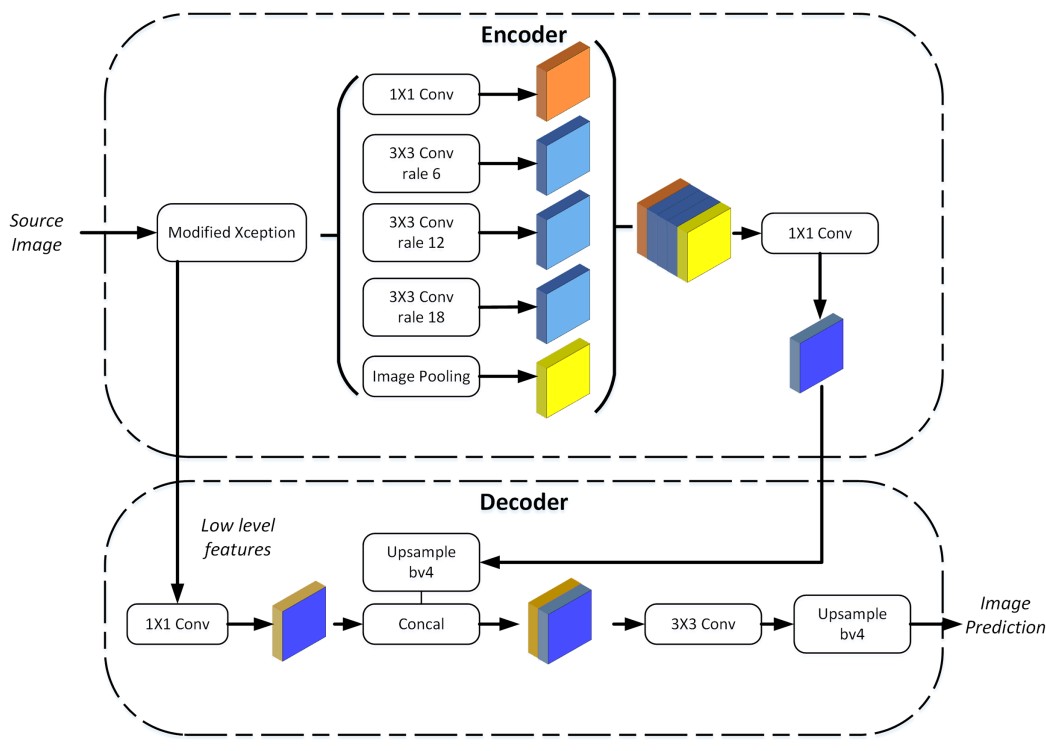

**Figure 2 The DeeplabV3+ network.**

### ASPP and codec

The Atrous Spatial Pyramid Pooling (ASPP) module aims to capture the contextual information of the input image and preserve spatial details. It mainly consists of multi-scale feature extraction and global pooling. The ASPP module extracts feature from the input feature map at different scales or receptive field sizes in the first part. Applying convolutions with different dilation rates allows the network to effectively capture multi-scale features without significantly increasing computation cost or reducing spatial resolution. This enables the network to perceive objects of different sizes within the input image. The second part involves global pooling on the input feature map. Global pooling aggregates information across the entire spatial dimensions of the feature map, typically using operations like average pooling or max pooling. This pooling operation helps the network capture global context information, which is beneficial for semantic understanding and improving segmentation accuracy. The ASPP module essentially consists of a $1 \times 1$ convolution (leftmost green block) + pooling pyramid (middle three blue blocks) + ASPP pooling (rightmost three layers), as shown in the diagram. The dilation factors of the pooling pyramid layers can be customized, allowing for flexible multi-scale feature extraction.

The codec module typically consists of an encoder and a decoder, specifically designed for image processing tasks. Its purpose is to improve the efficiency of both forward and backward propagation and reduce memory usage. In the encoder part, convolutional layers with learnable filters are used to extract hierarchical features from the input image. These filters are trained through backpropagation during the network's training phase. The

encoder includes multiple convolutional and pooling layers to gradually reduce the spatial dimensions of the feature map and extract high-level features. The input image is transformed into a lower-resolution feature map, preserving important information by applying successive convolution and pooling operations. The decoder part is responsible for gradually restoring the spatial resolution of the segmented image. This is achieved through up-sampling (*e.g.*, deconvolution or interpolation) and convolution. The decoder gradually recovers the low-resolution feature map to the same size as the original input image, effectively restoring spatial details. This process ultimately generates accurate segmentation results.

The combination of the ASPP module and the codec module in a deep learning architecture enhances the network's performance and efficiency for tasks like semantic segmentation. The ASPP module captures contextual information and spatial details, while the codec module improves the efficiency of information processing and memory usage, leading to more accurate segmentation and effective handling of large-scale tasks.

### Extended convolution and deep separable convolution

Expanded convolution (*Yu & Koltun, 2015*) injects holes into the core of standard convolution to have a large receptive field without passing through the pooling layer and aggregate a more comprehensive range of feature information without reducing the resolution. As shown in Fig. 3, the convolution core of $3 \times 3$ is an example to illustrate the increase of the receptive field. The receptive field of a $3 \times 3$ convolution nucleus with a void rate of two has increased to $7 \times 7$. Similarly, $3 \times 3$ convolution with a void rate of four can reach $15 \times 15$.

Depth separable convolution (*Chen et al., 2018*) decomposes the standard convolution into depth convolution and point-by-point convolution (Fig. 4), in which depth convolution does spatial convolution for each input channel independently point-by-point convolution is used to combine depth convolution output. Deep-separable volumes actively inhibit the increase of model parameters. DeeplabV3+ network will expand the separation convolution and apply it to ASPP and decoder modules.

### Effect of semantic analysis

We remove feature points that belong to dynamic pixels. We set the number of instances to six, including standard or potentially dynamic objects in the dataset, and mark them with different colors. Specifically, we set the human, chair, and monitor to light red, red, and blue. The result shows that each pixel in the image (*Sturm et al., 2012*) is assigned to different semantic categories. Humans are segmented into light red, indicating the areas where human motion occurs. Computers, tables, chairs, and the background are segmented into various colors, representing different objects or backgrounds in these areas. Through this semantic segmentation approach, a better understanding of the content in the image can be achieved, providing more accurate information for subsequent computer vision tasks.
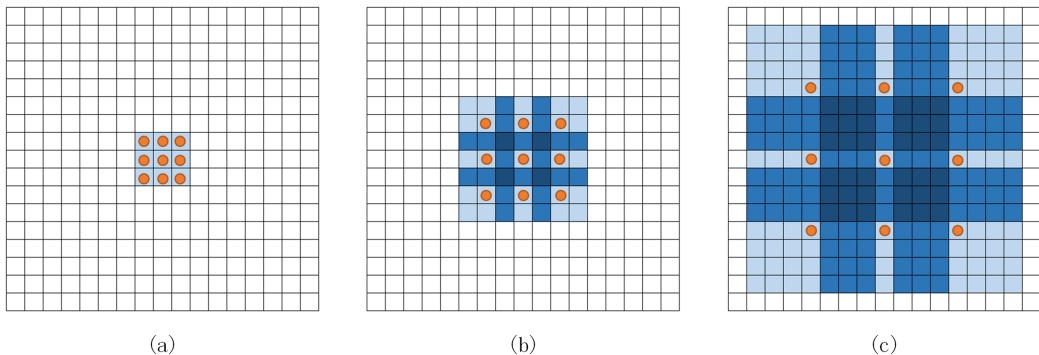

**Figure 3 Expanded convolution.** (A) Standard convolution. (B) 3 × 3 convolution with a void rate of 2. (C) 3 × 3 convolution with void rate of 4.

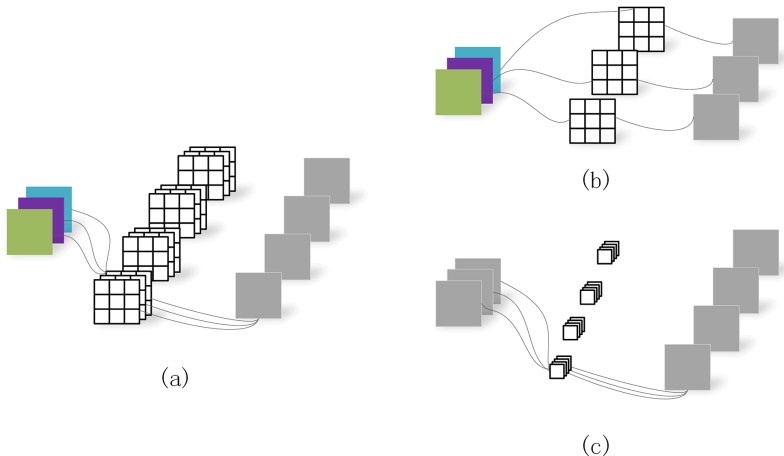

**Figure 4 Depth separable convolution.** (A) Standard convolution. (B) Depth convolution. (C) Depth separable convolution.

## Motion consistency detection

Through the DeeplabV3+ network, most dynamic objects can be segmented. However, the segmentation effect of potentially dynamically moving objects is not ideal, such as a book carried by a person or a chair moving with a person. To solve this problem, this article further checks whether the feature is dynamic by using the geometric constraint of the polar geometric feature. It will meet the polar constraint if it is a dynamic feature; otherwise, it is not.

The principle is as follows: In two consecutive frames of an image, the characteristic points $P_1$ and $P_2$. are the optical centers and cameras, respectively. The line connects and is called the baseline. The baseline and point determine a plane $\pi$ called the antipolar plane. The intersection lines of the plane $\pi$ and planes are called poles. The intersection points of the baseline and image plane are called opposite poles. The homogeneous coordinates of feature points can be expressed as Eq. (1).

$$P_1 = [u_1, v_1, 1], P_2 = [u_2, v_2, 1] \tag{1}$$

In the formula, $u_i, v_i (i = 1, 2)$ are the horizontal and vertical coordinates of $P_i(i = 1, 2)$. Baseline $l_1$ can be calculated by Eq. (2).

$$l_1 = \begin{bmatrix} X \\ Y \\ Z \end{bmatrix} = FP_1 = F \begin{bmatrix} u_1 \\ v_1 \\ 1 \end{bmatrix} \tag{2}$$

In the formal, $F$ is the basic matrix, which describes the mapping of the polar constraint from a point in one image to the corresponding polar line in another image. The mapping relationship can be described by Eq. (3).

$$P_2^T FP_1 = 0 \tag{3}$$

If the point $P_1$ and the base matrix $F$ are known, $P_2$ must satisfy the constraint if the space point $P$ is static. However, feature points with errors near the polar line are generated due to the uncertainty in feature extraction and basic matrix $F$ estimation. The two spatial point mapping image points do not strictly meet Eq. (3). As shown in Fig. 5B, point $P_2$ is not completely on the polar line but is very close to it. Therefore, the distance between point $P_2$ and polar line $l_2$ can be calculated by Eq. (4).

$$D = \frac{|P_2^T FP_1|}{\sqrt{\|X\|^2 + \|Y\|^2}} \tag{4}$$

If $D$ is less than the predetermined threshold, point $P$ is considered static; otherwise, it is considered dynamic.

The motion consistency detection algorithm is illustrated in Fig. 6. Dynamic features are detected using epipolar geometry constraints. First, the current set of matching feature points $l_2$ is computed using optical flow based on the previous frame feature point set $l_1$. If the matching pairs are too close to the image edge or if the pixel differences of the $3 \times 3$ image block in the center of the matching pairs are too significant, the matching pairs will be discarded. Then, at least five pairs of features are used to estimate the fundamental matrix $F$, usually using the classical eight-point algorithm. Finally, the current frame's epipolar lines are calculated using the fundamental matrix $F$. The distance between point $P_2$ and its corresponding epipolar line in frame $P_1$ is compared with a predetermined threshold to determine whether the feature point has moved.

The algorithm flow of motion consistency detection is shown in Fig. 6, which uses epipolar geometry constraints to detect dynamic features. Firstly, based on the feature point set $L_1$ of the previous frame, the matched feature point set $L_2$ in the current frame is calculated using optical flow. If the matching pairs are too close to the edge of the image, or if the pixel difference of the $3 \times 3$ image block at the center of the matching pairs is too significant, the matching pairs will be discarded. After that, at least five pairs of features can be used to estimate the fundamental matrix $F$, usually using the classic eight-point method, and then using the fundamental matrix $F$ to calculate the epipolar line in the current frame. Finally, it is determined whether the feature points have moved by calculating the distance between the corresponding epipolar lines of $P_2$ and $P_1$ and the preset threshold.

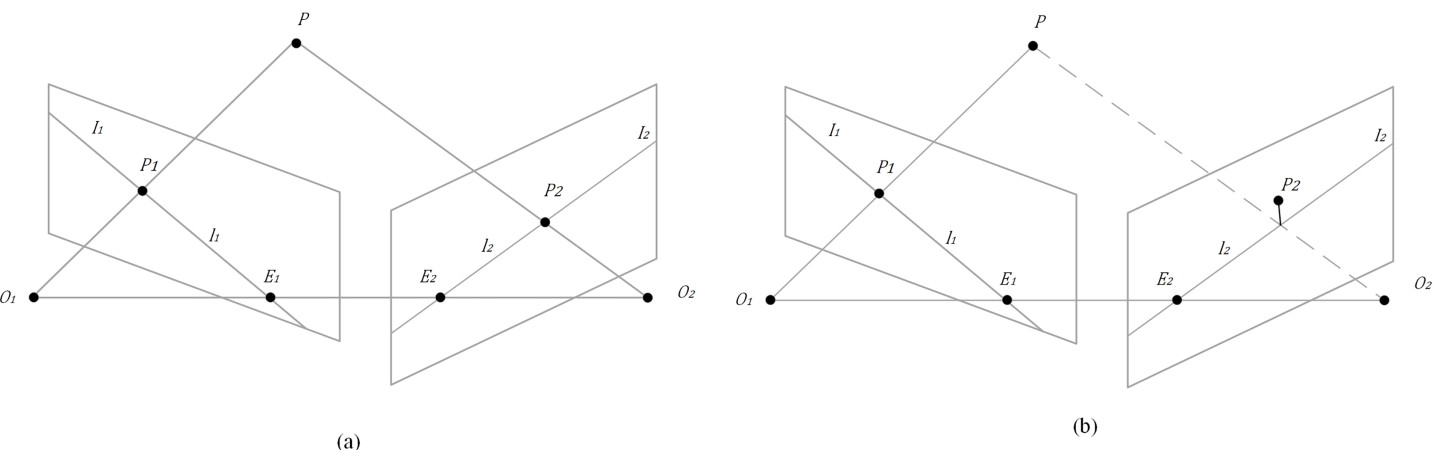

**Figure 5 Epipolar geometry.** (A) Correspondence between static points and feature points in consecutive adjacent frames. (B) Characteristic points with errors near the polar line.

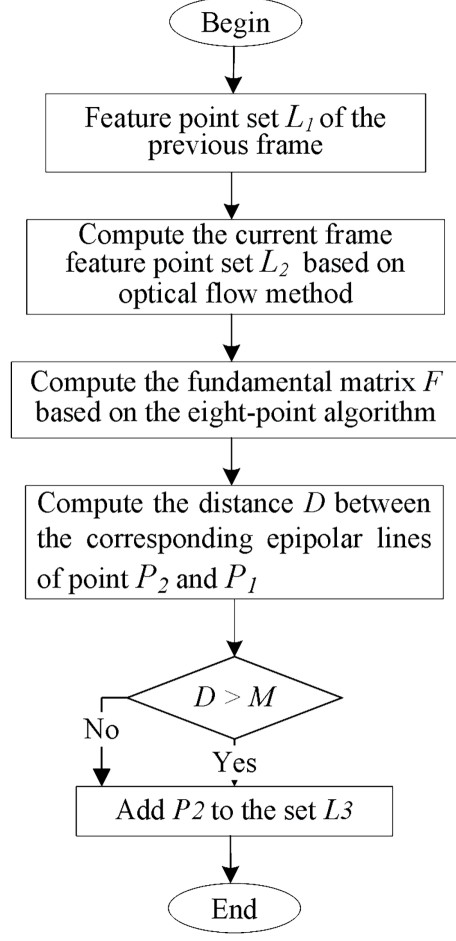

**Figure 6 Moving consistency check algorithm.**

## RESULTS

We compare three SLAM systems with the same dataset. Experiments show that the algorithm significantly improves the accuracy and robustness of the algorithm in a dynamic environment based on the original ORB-SLAM2. In order to evaluate the comprehensive ability of the system, the TUM public data set is used to conduct experiments on the system. The TUM data set consists of 39 sequences, which are recorded by Microsoft's Kinect sensor at a frequency of 30 Hz in different indoor scenes, including color images, depth images, and true poses, and contains texture-rich office dynamic scenes. The high dynamic sequence and low dynamic sequence meet the conditions for evaluating the comprehensive ability of the system in dynamic scenarios. At the same time, the data set also provides two standards for evaluating the tracking results of the SLAM system, namely absolute trajectory error and relative pose error. The absolute trajectory error directly calculates the difference between the camera pose's real value and the SLAM system's estimated value. This standard can very intuitively reflect the algorithm's accuracy and the trajectory's global consistency. The absolute trajectory error of the i-th frame is defined as in Eq. (5).

$$M_i := Q_i^{-1}ST_i \tag{5}$$

$Q_i$ is the algorithm-estimated pose of the i-th frame; $Q_i$ is the real pose of the i-th frame; $S$ is the transformation matrix from the estimated pose to the real pose. The relative pose error is used to calculate the difference between the real pose and the estimated pose within the same period $\Delta$. This standard is conducive to the evaluation of translation and rotation drift. The relative pose error of the i-th frame is defined in Eq. (6).

$$N_i := \left(Q^{-1}Q_{i+\Delta}\right)^{-1}\left(T_i^{-1}T_{i+\Delta}\right) \tag{6}$$

This experiment runs the traditional ORB-SLAM2 and the improved system on the above data sets. The computer configuration of the experiment is CPU i7-10750H, GPU RTX3080 16G. We conducted two experiments. The TUM RGB-D dataset fr3 is a widely used benchmark dataset for indoor visual SLAM. It contains various indoor scenes with different levels of complexity, including cluttered rooms, corridors, and offices. The dataset comprises synchronized RGB and depth image sequences and ground-truth poses obtained through a motion capture system. The dataset also includes many dynamic objects, with humans being the primary dynamic objects in the scenes. The dataset provides a challenging testbed for SLAM algorithms to handle various dynamic objects in the scene while maintaining accurate tracking and mapping.

## TRAJECTORY AND POSE ERRORS

We compared our system with the ORB-SLAM2 (*Mur-Artal & Tardos, 2017*) system and the Dynamic-SLAM (*Xiao et al., 2019*) system and evaluated them quantitatively using absolute trajectory and relative pose errors. The TUM dataset is a highly authoritative RGB-D indoor open-source dataset, where the dynamic scene part is widely used for various metric evaluations in the field of dynamic SLAM. The dynamic part are divided

into two major categories: low-dynamic scenes and high-dynamic scenes. The "TUM_fr3_sitting" sequence contains subtle movements of people and objects, while the "TUM_fr3_walking" sequence involves significant movements of people and objects, putting a great deal of strain on the tracking robustness of the system. The camera motions in these two categories can also be further divided into four subcategories: "xyz" denotes camera movements along the three main axes; "static" indicates minimal camera movement; "rpy" represents camera rotations about the roll, pitch, and yaw angles along the three axes; "half-sphere" refers to camera movements on a half-sphere. These different types of camera motions and scene variations provide a diverse and challenging dataset for evaluating the performance of dynamic SLAM systems under different conditions.

The experiment mainly used the walking sequence in the TUM dataset because people in the sequence move back and forth, which can be considered highly dynamic objects. The experiment also tested the sitting sequence, where people sit on chairs and move slightly, and can be considered as low dynamic objects. *xyz*, static, *rpy*, and half represent four types of camera ego-motion. We provide root mean square error (RMSE), mean error, median error, standard deviation, maximum error, and minimum error. RMSE and standard deviation can better reflect the stability and robustness of the system. As shown in the Fig. 7, the relative pose error graph for the high-dynamic sequence "*fr3_walking_xyz*" were compared between ORB-SLAM2, our system, and Dynamic_slam with the ground truth trajectory. Further comparison in Fig. 8 shows that the absolute trajectory error and relative pose error of the system described in the article are significantly smaller than those of ORB-SLAM2 and the Dynamic-SLAM system. This further validates that the article's semantic visual SLAM system, based on DeepLabV3+ for semantic segmentation and motion consistency detection, performs better by removing dynamic objects. Tables 1 and 2 present the typical value of absolute trajectory error and relative pose error respectively.

We evaluated the performance of our proposed method on an indoor scene dataset with dynamic objects. Our method achieved an average absolute translation error of 0.002 m and an average absolute rotation error of 2.099, which is an improvement over the baseline ORB-SLAM2 method, which had errors of 0.02497 m and 2.11 for the two metrics, respectively. In addition, our method successfully removed dynamic points from the feature set, resulting in more robust and accurate pose estimation. We further analyzes the performance of our method in scenes with varying degrees of dynamic objects.

Translation drift and rotational drift refer to the phenomena in which a robot or system gradually deviates from the expected path during the localization and mapping process. The accumulation of position estimation causes translational drift is inferred based on sensor data and motion models. However, inaccuracies in the sensors, motion model approximations, and unknown environment complexities can result in positioning errors. These errors accumulate over time, leading to a deviation between the estimated and actual positions, known as translational drift. On the other hand, rotational drift is typically caused by errors in the robot's orientation estimation. In SLAM, the robot must estimate its orientation or pose (*e.g.*, Euler angles or quaternions) to determine its positioning reletative to the environment. However, sensor errors, approximations in the motion

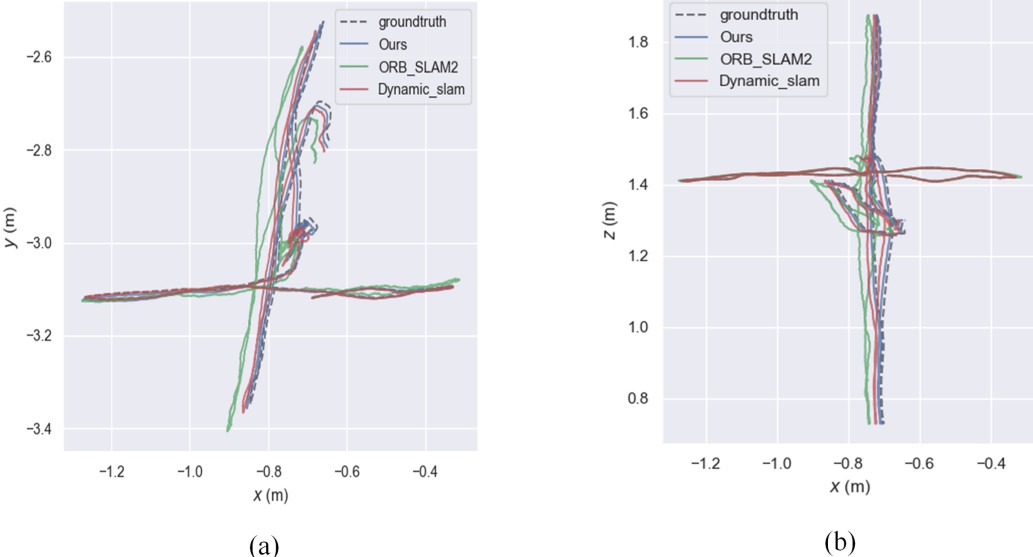

**Figure 7 Absolute trajectory error.** (A) Plotting the trajectory on the xy-plane. (B) Plotting the trajectory in the xz-plane.

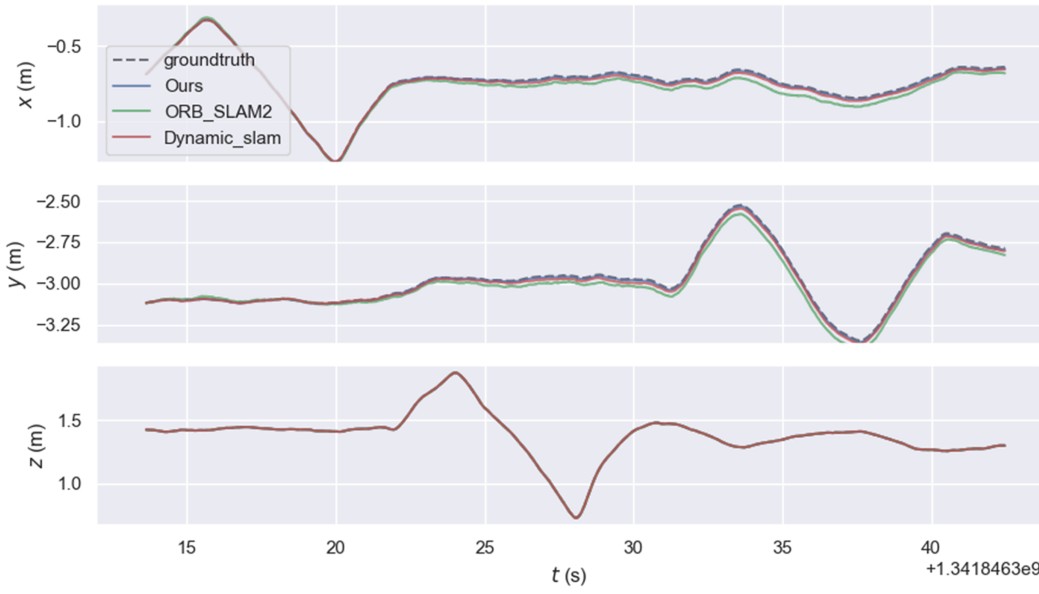

**Figure 8 Typical value of absolute trajectory error.**

models, and environmental uncertainties can introduce biases in the orientation estimation, leading to rotational drift.

Tables 3–5 presents the results of performance improvements in different types of sequences for a certain system. The improvements are measured in terms of Root Mean Square Error and standard deviation error. It seems that the system shows significant enhancements in more dynamic sequences, as indicated by the high improvement values in RMSE and standard deviation error. For example, in the fr3_walking_static sequence, the system achieved a remarkable 92.85% and 92.24% improvement in standard deviation

**Table 1 Typical value of absolute trajectory error.**

| Method | Parameter | Ours | ORB-SLAM2 | Dynamic_SLAM |
|---|---|---|---|---|
| APE w.r.t. translation part (m) | Max | 0.007 | 0.056 | 0.022 |
| | Mean | 0.003 | 0.026 | 0.009 |
| | Median | 0.003 | 0.024 | 0.009 |
| | RMSE | 0.003 | 0.03 | 0.01 |
| | SSE | 0.039 | 2.743 | 0.324 |
| | STD | 0.001 | 0.016 | 0.005 |
| APE w.r.t. rotation part (unit-less) | Max | 2.258 | 2.27 | 2.263 |
| | Mean | 2.096 | 2.11 | 2.101 |
| | Median | 2.099 | 2.113 | 2.104 |
| | RMSE | 2.098 | 2.112 | 2.103 |
| | SSE | 12,694.4 | 12,866.4 | 12,755 |
| | STD | 0.081 | 0.08 | 0.081 |

**Table 2 Typical value of relative pose error.**

| Method | Parameter | Ours | ORB-SLAM2 | Dynamic_SLAM |
|---|---|---|---|---|
| APE w.r.t. translation part (m) | Max | 0.01 | 0.011 | 0.01 |
| | Mean | 0.0025 | 0.0027 | 0.0026 |
| | Median | 0.0024 | 0.0025 | 0.0023 |
| | RMSE | 0.0031 | 0.0033 | 0.0032 |
| | SSE | 0.029 | 0.031 | 0.029 |
| | STD | 0.001 | 0.001 | 0.001 |
| APE w.r.t. rotation part (unit-less) | Max | 2.18 | 2.18 | 2.18 |
| | Mean | 0.0057 | 0.0059 | 0.0057 |
| | Median | 0.004362 | 0.004361 | 0.004362 |
| | RMSE | 0.041 | 0.046 | 0.041 |
| | SSE | 4.888 | 4.888 | 4.888 |
| | STD | 0.040773 | 0.040772 | 0.040778 |

error. This reflects the system's improved stability and robustness in handling such dynamic scenarios. However, traditional ORB-SLAM2 systems, known for their ability to handle low dynamic scenes effectively, already perform well in low dynamic sequences like the fr3_sitting_static sequence, leaving less room for further improvement with this system. In summary, the system showed significant enhancements in handing more dynamic sequences, but its performance was limited in scenarios with complex camera motions and did not bring substantial improvements in low dynamic scenes where traditional ORB-SLAM2 systems already perform well.

**Table 3 Translational drift.**

| Parameter | Data list | fr3_walking_static | fr3_walking_rpy | fr3_walking_helf | fr3_sitting_static |
|---|---|---|---|---|---|
| RMSE (m) | ORB-SLAM2 | 0.014 | 0.009 | 0.015 | 0.017 |
| | Ours | **0.001** | **0.002** | **0.004** | **0.001** |
| Mean (m) | ORB-SLAM2 | 0.0129 | 0.0076 | 0.0135 | 0.0146 |
| | Ours | **0.001** | **0.0018** | **0.0038** | **0.0015** |
| Median (m) | ORB-SLAM2 | 0.0114 | 0.0066 | 0.0119 | 0.0128 |
| | Ours | **0.0009** | **0.0014** | **0.0031** | **0.0013** |
| STD (m) | ORB-SLAM2 | 0.0068 | 0.0048 | 0.0078 | 0.0086 |
| | Ours | **0.0006** | **0.0012** | **0.0025** | **0.0009** |
| Improvement (%) | RMSE | 92.85 | 77.77 | 73.33 | 88.82 |
| | Mean | 92.24 | 76.31 | 71.85 | 89.72 |
| | Median | 92.10 | 78.78 | 73.94 | 89.84 |
| | STD | 91.17 | 75.00 | 67.94 | 89.53 |

**Note:**
The bold text indicates the improvement results.

**Table 4 Rotational drift.**

| Parameter | Data list | fr3_walking_static | fr3_walking_rpy | fr3_walking_helf | fr3_sitting_static |
|---|---|---|---|---|---|
| RMSE (°) | ORB-SLAM2 | 7.234 | 13.203 | 13.002 | 0.3321 |
| | Ours | **1.61** | **8.49** | **3.12** | **0.32** |
| Mean (°) | ORB-SLAM2 | 4.768 | 13.29 | 9.203 | 0.312 |
| | Ours | **1.26** | **6.39** | **4.28** | **0.28** |
| Median (°) | ORB-SLAM2 | 0.626 | 11.39 | 8.39 | 0.31 |
| | Ours | **0.201** | **5.194** | **2.394** | **0.279** |
| STD (°) | ORB-SLAM2 | 6.5391 | 8.4947 | 6.0083 | 0.1342 |
| | Ours | **0.239** | **5.242** | **2.104** | **0.123** |
| Improvement (%) | RMSE | 77.74 | 35.69 | 76.00 | 3.64 |
| | Mean | 73.57 | 51.92 | 53.49 | 10.25 |
| | Median | 67.89 | 54.39 | 71.47 | 10.00 |
| | STD | 96.34 | 38.29 | 64.98 | 8.34 |

**Note:**
The bold text indicates the improvement results.

# DISCUSSION

In summary, ORB-SLAM2 is a powerful visual SLAM system that performs well in static or mildly dynamic environments. However, it has limitations when dealing with more complex dynamic scenes, primarily due to its reliance on feature point extraction and matching for localization and map construction. Feature points in dynamic scenes can become unstable or undergo frequent changes, leading to positioning errors and unstable map reconstruction. To address the challenges of dynamic scenes, researchers are making efforts to improve the robustness and adaptability of SLAM systems. Among various approaches, introducing semantic information shows promising prospects. By comprehending the semantics of objects in the scene, the SLAM system can better

**Table 5  Absolute trajectory error.**

| Parameter | Data list | fr3_walking_static | fr3_walking_rpy | fr3_walking_helf | fr3_sitting_static |
|---|---|---|---|---|---|
| RMSE (m) | ORB-SLAM2 | 0.2219 | 0.5612 | 0.2354 | 0.0054 |
|  | Ours | **0.008** | **0.29** | **0.072** | **0.004** |
| Mean (m) | ORB-SLAM2 | 0.2231 | 0.5452 | 0.2215 | 0.0031 |
|  | Ours | **0.005** | **0.254** | **0.055** | **0.002** |
| Median (m) | ORB-SLAM2 | 0.1622 | 0.5485 | 0.1214 | 0.0064 |
|  | Ours | **0.006** | **0.1334** | **0.0513** | **0.0052** |
| STD (m) | ORB-SLAM2 | 0.1365 | 0.2033 | 0.2531 | 0.0024 |
|  | Ours | **0.001** | **0.182** | **0.051** | **0.002** |
| Improvement (%) | RMSE | **96.39** | **48.33** | **69.41** | **25.92** |
|  | Mean | **97.75** | **53.34** | **75.16** | **35.48** |
|  | Median | **96.30** | **75.68** | **57.74** | **18.75** |
|  | STD | **98.60** | **10.47** | **79.84** | **16.67** |

**Note:**
The bold text indicates the improvement results.

distinguish between static and dynamic objects and exclude dynamic objects from the localization and map construction process, thus enhancing system stability. The introduction of semantic information can be achieved through deep learning techniques, and this article utilizes DeeplabV3+ to realize this goal.

We can address these limitations by incorporating DeepLabV3+, which utilizes dilated convolutions to increase the receptive field without increasing parameter count, and introducing the Atrous Spatial Pyramid Pooling (ASPP) module for feature extraction. The ASPP module captures semantic information at different scales through dilated convolutions with different sampling rates, improving the accuracy of segmentation results and filtering out dynamic objects. After being processed by DeepLabV3+, the motion consistency check algorithm is further applied to determine whether the remaining feature points belong to the static background or dynamic objects. The algorithm observes the motion patterns of feature points between adjacent frames: if the motion pattern is consistent with the surrounding features and matches the expected camera motion pattern, the feature points are considered part of the static background. Conversely, if the motion pattern of the feature point is inconsistent with the surrounding feature points or does not match the expected camera motion pattern, it is considered a dynamic object.

## CONCLUSIONS

Applying DeepLabV3+ to ORB-SLAM2 enhances the system's perception and semantic understanding capabilities of dynamic objects. Through the dual filtering process, feature points belonging to dynamic objects are effectively eliminated, thereby improving the robustness and accuracy of the SLAM system in dynamic environments. Compared with the traditional ORB-SLAM2 visual SLAM system, the prior semantic knowledge is obtained to provide an advanced understanding of the environment by introducing a deeplabv3+ network to detect potential dynamic objects. Combining the motion consistency detection algorithm with the semantic segmentation algorithm makes the

improved system significantly less in absolute trajectory error and relative position error than the traditional ORB-SLAM2 and dynamic environment, which improves the accuracy and robustness of the system position estimation. In future work, further optimization of the motion consistency detection algorithm could enhance the system's performance, potentially by incorporating more advanced techniques or leveraging additional sensor modalities. Furthermore, the system could be extended to handle more complex and challenging environments, such as crowded urban scenes or poor light environments, to evaluate its effectiveness in such scenarios.

### Funding

This work was supported by the Universiti Kebangsaan Malaysia under Grant Code: TT-2023-012 and GP-K021817. The funders had no role in study design, data collection and analysis, decision to publish, or preparation of the manuscript.

### Grant Disclosures

The following grant information was disclosed by the authors:
Universiti Kebangsaan Malaysia: TT-2023-012 and GP-K021817.

### Competing Interests

The authors declare that they have no competing interests.

### Author Contributions

- Xiao Ya Zhang conceived and designed the experiments, performed the experiments, analyzed the data, performed the computation work, prepared figures and/or tables, authored or reviewed drafts of the article, and approved the final draft.
- Abdul Hadi Abd Rahman conceived and designed the experiments, performed the experiments, analyzed the data, performed the computation work, authored or reviewed drafts of the article, and approved the final draft.
- Faizan Qamar analyzed the data, prepared figures and/or tables, authored or reviewed drafts of the article, and approved the final draft.

### Data Availability

The code is available at GitHub and Zenodo:
- https://github.com/xiaoyayazhang/deeplabv3-vslam
- ZHANG, Abd Rahman, & Qamar. (2023). deeplabv3+slam (Version 1). Zenodo. https://doi.org/10.5281/zenodo.8249278
The datasets are available at TUM School of Computation, Information and Technology, under Dynamic Objects: https://cvg.cit.tum.de/data/datasets/rgbd-dataset/download
- fr3_walking_xyz
- fr3_walking_static

- fr3_walking_rpy
- fr3_walking_helf
- fr3_sitting_static

Dataset licence: Unless stated otherwise, all data in the TUM RGB-D benchmark is licensed under a Creative Commons 4.0 Attribution License (CC BY 4.0) and the accompanying source code is licensed under a BSD-2-Clause License (https://cvg.cit.tum.de/data/datasets/rgbd-dataset).

## Supplemental Information

Supplemental information for this article can be found online at http://dx.doi.org/10.7717/peerj-cs.1628#supplemental-information.

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
