# Peer review of "Semantic visual simultaneous localization and mapping (SLAM) using deep learning for dynamic scenes"

_PeerJ Computer Science, doi:10.7717/peerj-cs.1628_

## Round 0.1 · original submission · Major Revisions

· Academic Editor

Major Revisions

All reviewers have recommended major revisions for this article. The authors need to prepare revisions and rebuttal carefully considering all issues highlighted by the reviewers.

Reviewer 1 ·

Basic reporting

1. There are few grammatical errors in the paper.
2. The introduction part of the paper needs to be improved as it doesn't demonstrate how their work fits into the broader field of knowledge.

Experimental design

Experimental design of the study is somehow ok but it needs more refinement specially with the computational complexity of their algorithm. How their algorithm is computationally less complex?

Validity of the findings

No Comment

Additional comments

The paper presents an approach to enhance the robustness and precision of monocular visual odometry in dynamic environments. However, there are several concerns:
1. In the abstract, the authors mention the computational complexity and uncertainties in data association, but the paper does not explain how they addressed the computational complexity or how their algorithms are less computationally complex.
2. The authors state that the current research trend involves using deep learning-based methods to improve accuracy, but they fail to compare their work with other relevant studies.
3. The paper only compares the authors' approach with the traditional ORB-SLAM2 systems and dynamic SLAM systems.
4. The authors claim that their approach outperforms the baseline ORB-SLAM2 with an RMSE improvement of up to 83.89% and assert that their results demonstrate a significant improvement. However, these claims lack support from a comparative study with similar works.
5. The authors mention in Line 188 that matched features are used for estimating the camera's pose through triangulation, but the paper does not provide details on how triangulation assists with the estimation or which specific triangulation method the authors employ.

·

Basic reporting

Manuscript entitled “Semantic Visual SLAM using Deep Learning for Dynamic Scenes”, addresses the one of common issues involved with vision SLAM approach. Authors proposed an approach for improving the accuracy of traditional SLAM approach by eliminating the dynamic feature point, which one of the main cause in the error accumulation. In the proposed method well known semantic segmentation approach Deeplab V3+ used to identify the dynamic object in the scene. And after identifying, all the feature points of dynamic object removed and only static feature considered for pose estimation. And experimental analysis conducted on benchmark dataset to prove the significance of proposed method.

Manuscript written well with good editing standards and sentence ambiguity, grammar error relatively rare. Although study appears to be interesting in automation filed, there are already similar work has been discussed in the published articles. There is nothing new in the objective and research finding highlighted in the manuscript. For reference, please check following few article. Moreover it is well-known issue that, visual and lidar odometry algorithms performance negatively impacted by dynamic object in the scene and there are various approach introduced to successfully address this issue.

Experimental design

There are few similar approach found in the literature used different object detection models to eliminate the feature points of dynamic objects. In this manuscript authors choose Deeplab V3+ detection module for identifying the dynamic object in the scene. But it is necessary to discuss, clear motivation and significant advantages found by authors by using Deeplab V3+ model. Why not other model ?

Validity of the findings

Core contribution of this proposed work is relies on, how efficiently proposed approach able to differentiate dynamic feature points from static feature points. More specifically, Section Motion consistency detection discuss in detail. But it is necessary to present some result images depicting the actual process. With typical object detection module used for motion detection and proposed motion consistency algorithm out on same scene.

More importantly, as authors mentioned in the abstract proposed approach outperforms traditional visual odometry in terms of accuracy and robustness. It is very necessary to provide the evidence for robustness with proper time compassion analysis.
Also, proposed method should be compared with existing approach which addressed similar issues in the literature. Only comparing with typical approach may not be appropriate.

Additional comments

1. Figure 2, represents the Deeplabv3+ network architecture, showing input image with cat. Though it is meaningful illustration to describe the proposed SLAM approach. I would strongly recommend modify with the scene (instead cat), more commonly appear in the indoor SLAM.
2. In the Motion consistency detection section, Line# 303, refers Fig 6 mentioning motion consistency detection algorithm illustration. But actual image is number is Fig 7. Please check image numbers before referencing.
3. Maintain the consistency in using phrase, it is used that Motion consistency at description, but image subtitle written as Moving consistency.
4. Check Fig8, Image subtitles (d), (e) and (f) repeated.
5. Check Cap and Small letter consistency of Subsection heading entire manuscript. It is randomly used.
Extended convolution and deep separable convolution
Effect of Semantic Analysis
Motion consistency detection
Feature matching
Trajectory and Pose Errors
6. Table 4, Rotational drift analysis presents observed rotational error. But units mentioned as (m), Suggest authors to mention the unit appropriately.

Reviewer 3 ·

Basic reporting

The article presents a clear overview of the problem being addressed and the proposed approach to enhance monocular visual odometry in dynamic environments. The use of semantic segmentation with DeeplabV3+ to identify dynamic objects and then apply motion consistency check to remove feature points belonging to dynamic objects is a promising idea to improve accuracy and robustness.

=> Suggested to change "Related Work" section to "Background".

Experimental design

No comment

Validity of the findings

1. The discussion section in the article provides a clear and comprehensive overview of the limitations of ORB-SLAM2 in dynamic environments and proposes a potential solution using DeepLabV3+ and a motion consistency check algorithm. However, it would be beneficial to include specific quantitative results or comparative analyses to substantiate the claims made about the limitations of ORB-SLAM2 in dynamic environments. Without such evidence, it is challenging to fully assess the extent of the problem and the potential improvements achieved by the proposed approach.

2. Additionally, the authors could elaborate on any specific challenges or potential issues that might arise when applying the motion consistency check algorithm in different dynamic environments, as different scenarios may require adaptive or fine-tuning approaches.

3. The conclusion provides a clear and concise summary of the main findings and improvements achieved by incorporating DeepLabV3+ into ORB-SLAM2. However, to strengthen the conclusion, it would be beneficial to include specific quantitative results and statistical analysis comparing the performance of the improved system with the traditional ORB-SLAM2 in dynamic environments. This would provide more solid evidence to support the claims of enhanced accuracy and robustness.

Additional comments

1. It is suggested to update all the figures with good-quality images.
2. Correct the numbering (vi => iv) in line 54 (SLAM advantages).
3. Define p in line 294 (if p is static).
4. Cross-check the line 297, p2 is not represented in Fig 9. "As shown in Fig.9 (b), point ÿ is not completely on the polar line p2"
5. Verify and correct the figure numbers in line 360 (Fig.9d, 9e, and 9f). There are no such figures numbers in this article.
6. Check line 252 and 253, "rate of 2" and "ratio of 4".

---

## Round 0.2 · accepted · Accept

· Academic Editor

Accept

The authors have revised the article to address all the concerns of the reviewers.

Reviewer 1 ·

Basic reporting

No Comment

Experimental design

No Comment

Validity of the findings

No Comment

Additional comments

No Comment

Reviewer 3 ·

Basic reporting

I have no more comments.

Experimental design

No Comments

Validity of the findings

Thanks for all your clarifications and correction.

Additional comments

I think this revised version has improved the previous one.